# Medicinal Chrysanthemum Detection under Complex Environments Using the MC-LCNN Model

**DOI:** 10.3390/plants11070838

**Published:** 2022-03-22

**Authors:** Chao Qi, Jiangxue Chang, Jiayu Zhang, Yi Zuo, Zongyou Ben, Kunjie Chen

**Affiliations:** 1College of Engineering, Nanjing Agricultural University, Nanjing 210031, China; chaoqi.njau@gmail.com (C.Q.); jiayuzhang.njau@outlook.com (J.Z.); yizuonjau@163.com (Y.Z.); ZYBen90@163.com (Z.B.); 2College of Intelligent Engineering and Technology, Jiangsu Vocational Institute of Commerce, Nanjing 211168, China; cjxsnow@126.com

**Keywords:** chrysanthemum, bud stage detection, deep convolutional neural network, agricultural robotics, edge computing device

## Abstract

Medicinal chrysanthemum detection is one of the desirable tasks of selective chrysanthemum harvesting robots. However, it is challenging to achieve accurate detection in real time under complex unstructured field environments. In this context, we propose a novel lightweight convolutional neural network for medicinal chrysanthemum detection (MC-LCNN). First, in the backbone and neck components, we employed the proposed residual structures MC-ResNetv1 and MC-ResNetv2 as the main network and embedded the custom feature extraction module and feature fusion module to guide the gradient flow. Moreover, across the network, we used a custom loss function to improve the precision of the proposed model. The results showed that under the NVIDIA Tesla V100 GPU environment, the inference speed could reach 109.28 FPS per image (416 × 416), and the detection precision (AP_50_) could reach 93.06%. Not only that, we embedded the MC-LCNN model into the edge computing device NVIDIA Jetson TX2 for real-time object detection, adopting a CPU–GPU multithreaded pipeline design to improve the inference speed by 2FPS. This model could be further developed into a perception system for selective harvesting chrysanthemum robots in the future.

## 1. Introduction

Numerous studies have reported that medicinal chrysanthemums have significant commercial value [1]. Furthermore, it has prominent medicinal values [2], such as heat clearing, eye brightening, anti-inflammatory, antihypertensive, and antitumor properties. In the natural environment, a single chrysanthemum plant can present flower heads in different flowering stages, whereas medicinal chrysanthemums are mainly harvested at the bud stage. To show the research objective of this study, the different flowering stages of medicinal chrysanthemums are presented in Figure 1.

At present, the harvesting process of medicinal chrysanthemums is labor-intensive and time-consuming. Consequently, due to the current shortage of skilled labor, it is highly desirable to develop a selective harvesting robot to solve the crop waste problem. The design of manipulators and the development of visual perception systems are vital for selective harvesting robots, and this study is focused on the development of visual perception systems for medicinal chrysanthemums. Traditional machine learning techniques for computer vision tasks are well developed, with shallow learning of image information through manual feature extraction [3]. Convolutional neural networks (CNNs), an important subset of machine learning techniques that learn hierarchical representations and discover potentially complex patterns from the data, have made impressive advances in the computer vision field [4]. CNNs have also yielded encouraging results in agriculture [5]. Although the approaches based on traditional machine learning techniques and deep learning techniques have achieved significant success in agricultural applications, developing lightweight networks for selective harvesting robots under unstructured environments is still difficult.

We collected the literature on chrysanthemum detection based on traditional machine learning techniques and deep learning techniques throughout the world, and the results are shown in Table 1. Overall, the available literature is relatively scarce. When carefully analyzing Table 1, we found three issues that deserve further exploration.

Issue 1: The current research has not yet achieved high-accuracy, real-time detection of chrysanthemums.

Issue 2: Throughout the literature, the testing environment has mainly been in the laboratory, which cannot guarantee the robustness of the model.

Issue 3: Although there are some differences in the research tasks for chrysanthemum detection, the aim of the research is to achieve commercialization, and this could be effective in helping farmers reduce their workload. Commercial production inevitably requires embedding the models into low-power edge computing devices, but the current test results are laptop-based.

Based on the three issues above, we propose a lightweight convolutional neural model (MC-LCNN). First, MC-LCNN can balance detection precision and inference speed to achieve real-time and efficient detection of medicinal chrysanthemums. Second, MC-LCNN was tested under three complex unstructured environments (illumination variations, overlaps, and occlusions) to ensure the robustness of the proposed model. Finally, for subsequent development of selective harvesting chrysanthemum robots, we chose to test MC-LCNN on a low-power embedded GPU platform, NVIDIA Jetson TX2, and further improved the detection inference speed by designing a CPU–GPU heterogeneous architecture. The contributions of this study are as follows:A lightweight MC-LCNN model was designed to achieve high-accuracy, real-time detection of medicinal chrysanthemums under complex unstructured environments.A series of experiments were designed to validate the superiority of MC-LCNN, including comparisons with different data enhancements, ablation experiments between various network components, and comparisons with state-of-the-art object detection models.The MC-LCNN model was embedded into an edge computing device with a custom pipeline design to achieve accurate real-time medicinal chrysanthemum detection.

The rest of the paper is organized as follows. Section 2 describes the dataset, the hardware parameters of the NVIDIA Jetson TX2, the structure of the proposed model, the improvement approach of multithreading, the evaluation metrics, and the experimental setup. Section 3 presents the experimental results in detail. Section 4 discusses the experimental results, advantages and disadvantages, solutions, and future research perspectives of this work. Section 5 briefly summarizes the contributions of this study.

## 2. Materials and Methods

### 2.1. Dataset

The medicinal chrysanthemum dataset used in this study was collected at Yangma Town, China, from October 2019 to October 2021. Due to the short flowering stage of medicinal chrysanthemums, there are only a few days per year to collect suitable samples. The capture device was an Apple X phone with a video resolution of 1080 × 1920. The dataset was collected entirely in the field, with backgrounds including illumination variations, occlusions, and overlaps. It is worth mentioning that to ensure the robustness of the robotic perception system, the collected images had no natural environmental constraints. The dataset comprising a total of 4000 chrysanthemum images was divided into training, validation, and test datasets following a ratio of 6:3:1. Some original images are shown in Figure 2.

### 2.2. NVIDIA Jetson TX2

The NVIDIA Jetson TX2 comprises a 6-core ARMv8 64-bit CPU complex and a 256-core NVIDIA Pascal architecture GPU. The CPU complex includes a dual-core Denver2 processor, a quad-core ARM Cortex-A57, 8 GB of LPDDR4 memory, and a 128 bit interface, making it ideal for applications with low power and high computing performance. Therefore, we chose this edge computing device to design and implement a real-time object detection system. We introduce the NVIDIA Jetson TX2 in Figure 3.

### 2.3. MC-LCNN

The MC-LCNN is a lightweight network (11.3 M) that can achieve real-time detection of complex unstructured environments (light changes, occlusions, and overlaps). The network structure is mainly constructed based on the backbone, neck, and head, as shown in Figure 4. In the backbone, the main network utilizes the proposed MC-ResNetv1 incorporating the CBM module and SPP module in this component. In the neck, the main network uses the proposed MC-ResNetv2 with the CBL module embedded. In the head, a feature pyramid network (FPN) feature fusion strategy is employed. Furthermore, several strategies were used throughout the network to improve the training robustness, including exponential moving average (EMA), larger batch size, DropBlock regularization, and generalized focal loss.

#### 2.3.1. MC-ResNetv1 and MC-ResNetv2

The main challenge in implementing lightweight models is that under fixed computational budgets (FLOPs), only a restricted amount of feature channels can be afforded. To increase the number of channels at low computational budgets, we employed a 1 × 1 convolution and a bottleneck structure to achieve information exchange between different channels. The shape of the 1 × 1 convolution was determined by the input channels *c*_1_ and output channels *c*_2_. Thus, the FLOPs of the 1 × 1 convolution could be calculated as $B=hwc_{1}c_{2}$, where *h* and *w* are the spatial sizes of the feature maps. When the cache in the computing device is sufficiently large to store all the feature maps and parameters, the memory access cost $\left(mac\right)=hw\left(c_{1}+c_{2}\right)+c_{1}c_{2}$. Based on the mean inequality, we obtain the following:(1)$$\mathrm{mac}\;\geq 2\sqrt{hwB}+\frac{B}{hw}$$

Accordingly, the memory access cost has a minimum value given by the FLOPs. It reaches its minimum value when the number of input and output channels are equal.

A 1 × 1 convolution reduces the computational burden by replacing dense convolution with sparse convolution. On the one hand, it allows more channels to be used at fixed FLOPs and increases the network capacity. However, on the other hand, the increase in the number of channels leads to a higher memory access cost. The relationship between memory access cost and FLOPs for 1 × 1 group convolution is as follows:(2)$$\begin{array}{c} mac=hw\left(c_{1}+c_{2}\right)+\frac{c_{1}c_{2}}{g}\\ =hwc_{1}+\frac{Bg}{c_{1}}+\frac{B}{hw}, \end{array}$$
where g denotes the number of groups, and $B=hwc_{1}c_{2}/g$ stands for FLOPs. Given the fixed input shape $c_{1}\times h\times w$ and the computational cost B, the memory access cost increases with the growth of g.

Both 1 × 1 convolution and bottleneck structures increase the memory access cost. This cost is not negligible, especially for lightweight networks. Consequently, to obtain ze high model capacity and efficiency, the critical issue is how to keep numerous equal-width channels without either dense convolution or many groups. To achieve the above, we designed the MC-ResNetv1 module. We introduced a simple operator named Focus, where the input is split into two branches at the beginning of each unit. One branch uses a shortcut design, where half of the feature channels directly passes through the block and joins the next block, which can be considered as functional reuse. The other branch comprises two convolutions with the same input and output channels. Moreover, another MC-ResNetv2 module was designed, where the Focus operation was removed and thereby the number of output channels was doubled. At the same time, the original shortcut design was substituted with two convolutions. The blocks were repeatedly stacked to construct the entire network. Therefore, 3 × 3 convolutions are followed by an additional 1 × 1 convolutional layer to blend the features, and the number of channels in each block is scaled to generate a network of different complexities. Not only that, the 1 × 1 convolution removes computational bottlenecks by reducing the dimensionality of the module, which is otherwise constraining the size of the network. This not only increases the depth of the network but also increases the width of the network without significantly affecting performance. To verify the performance of MC-ResNetv1 and MC-ResNetv2, we implemented ablation experiments, as outlined in Section 3.2.

#### 2.3.2. Generalized Focal Loss

Focal loss [14] is designed for object detection tasks with an imbalance between the foreground and background classes, and Equation (3) is as follows:(3)$$\mathrm{FL}\;\left(p\right)=-{\left(1-p_{t}\right)}^{\gamma }\;\log\;\left(p_{t}\right),\;p_{t}=\{\begin{array}{ccc} p, & \mathrm{when} & y=1\\ 1-p, & \mathrm{when} & y=0 \end{array}$$
where y∈{1,0} denotes the ground truth class, p∈{1,0} indicates the estimated probability of the class labeled as y=1, and γ represents an adjustable focusing parameter. To be specific, focal loss comprises a dynamically scaling factor part ${\left(1-p_{t}\right)}^{\gamma }$ and a standard cross-entropy part $-\log\left(p_{t}\right)$. Due to the presence of class imbalance problems, we considered extending the two components of focal loss, known as the quality focus loss (*Q*): (4)$$\;Q\left(\sigma \right)=-|y-\sigma {|}^{\beta }\;\left(\left(1-y\right)\;\log\;\left(1-\sigma \right)+y\;\log\;\left(\sigma \right)\right)$$
where *σ* = *y* means the global minimum solution of the quality focus loss. $|y-\sigma {|}^{\beta }$ is a moderating factor that goes to 0 when the quality estimate becomes accurate, i.e., σ→y, and the loss of well-estimated samples is downgraded, where the parameter β smoothly controls the downgraded rate. We used the relative offset from the location to the four sides of the bounding box as the regression objective. The bounding box regression models the regression label y as Dirac delta distribution δ(x−y), where $\int_{-\infty }^{+\infty }\delta \left(x-y\right)dx=1$. The integral of y is as follows:(5)$$y=\int_{-\infty }^{+\infty }\delta \left(x-y\right)xdx$$

We learnt the underlying generic distribution P(x) directly without inserting any other prior factors instead of the Dirac delta assumption. Based on the range of labels for the minimum y0 and maximum y0 (y0≤y≤yn,n∈ℕ+), we can estimate y^ from the model:(6)y^=∫−∞+∞P(x)xdx=∫y0ynP(x)xdx

To be consistent with the network structure, we discretized the range [$y_{0}$,$y_{n}$] as a set of $\left\{y_{0},y_{1},\ldots ,y_{i},y_{i+1},\ldots ,y_{n-1},y_{n}\right\}$, converting the integral of the continuous domain into a discrete representation. Thus, according to the discrete distribution property $\sum_{i=0}^{n}P\left(y_{i}\right)=1$, the regression value $\hat{y}\;$can be formulated as follows:(7)$$\hat{y}=\sum_{i=0}^{n}P\left(y_{i}\right)y_{i}$$

Consequently, P(x) can be simply achieved by the softmax S(·) layer, where $P\left(y_{i}\right)$ is represented as $S_{i}$.

To encourage high probability values close to the target y to optimize P(x), we introduced a distribution focus loss. By expanding the probabilities of $y_{i}$ and $y_{i+1}$, the network is forced to concentrate quickly on values close to the label *y*. We defined the distributional focus loss by applying the entire cross-entropy component of the mass focus loss. We defined distribution focus loss by applying the whole cross-entropy part of quality focus loss:(8)$$\;Q\left(S_{i},S_{i+1}\right)=-\left(\left(y_{i+1}-y\right)\log\left(S_{i}\right)+\left(y-y_{i}\right)\log\left(S_{i+1}\right)\right)$$

The purpose of distribution focus loss is to expand the probability of the values around the target y. The global minimum solution of distribution focus loss, i.e., $S_{i}=\frac{y_{i+1}-y}{y_{i+1}-y_{i}},S_{i+1}=\frac{y-y_{i}}{y_{i+1}-y_{i}}$, ensures that the estimated regression target $\hat{y}$ is infinitely close to the corresponding label y, i.e., $\hat{y}=\sum_{j=0}^{n}P\left(y_{j}\right)y_{j}=S_{i}y_{i}+S_{i+1}y_{i+1}=\frac{y_{i+1}-y}{y_{i+1}-y_{i}}y_{i}+\frac{y-y_{i}}{y_{i+1}-y_{i}}y_{i+1}=y$.

Quality focus loss and distribution focus loss can be unified into a general form known as generalized focal loss. Suppose a model has probability estimates for two variables $y_{l},y_{r}\;\left(y_{l}<y_{r}\right)$ as $p_{y_{l}},p_{y_{r}}\;\left(p_{y_{l}}\geq 0,p_{y_{r}}\geq 0,p_{y_{l}}+p_{y_{r}}=1\right)$, and the final prediction of their linear combination is $\hat{y}=y_{l}p_{y_{l}}+y_{r}p_{y_{r}}\left(y_{l}\leq \hat{y}\leq y_{r}\right)$. The corresponding label y of the predicted $\hat{y}$ also satisfies $y_{l}\leq y\leq y_{r}$. With the absolute distance $|y-\hat{y}{|}^{\beta }\left(\beta \geq 0\right)$ as the moderating factor, the equation of generalized focal loss (*G*) is as follows:(9)$$\;G\left(p_{y_{l}},p_{y_{r}}\right)=-{\left|y-\left(y_{l}p_{y_{l}}+y_{r}p_{y_{r}}\right)\right|}^{\beta }\left(\left(y_{r}-y\right)\log\left(p_{y_{l}}\right)+\left(y-y_{l}\right)\log\left(p_{y_{r}}\right)\right)$$

Generalized focal loss $\left(p_{y_{l}},p_{y_{r}}\right)$ reaches a global minimum at $p_{y_{l}}^{*}=\frac{y_{r}-y}{y_{r}-y_{l}}$ and $p_{y_{r}}^{*}=\frac{y-y_{l}}{y_{r}-y_{l}}$, which also implies that $\hat{y}$ exactly matches the continuous label y, i.e., $\hat{y}=y_{l}p_{y_{l}}^{*}+y_{r}p_{y_{r}}^{*}=y$. The modified detector differs from the former detector in two respects. First, we fed the classification scores directly as NMS scores during the inference process without multiplication if any separate quality prediction existed. Second, the final layer of the regression branch used to predict the location of each bounding box now has n+1 outputs rather than l output, resulting in negligible additional computational cost. We can define the training loss ℒ in terms of generalized focal loss as follows:(10)$$ℒ=\frac{1}{N_{\mathrm{pos}\;}}\sum_{z}{ℒ}_{Q}+\frac{1}{N_{\mathrm{pos}\;}}\sum_{z}1_{\left\{c_{z}^{*}>0\right\}}\left(\lambda_{0}{ℒ}_{ℬ}+\lambda_{1}{ℒ}_{D}\right)$$
where ${ℒ}_{Q}$ is quality focus loss, and ${ℒ}_{D}$ is distribution focus loss. ${ℒ}_{ℬ}$ stands for GIoU loss, and $\lambda_{0}$ and $\lambda_{1}$ refer to the balance weights of ${ℒ}_{Q}$ and ${ℒ}_{D}$, respectively. Here, $1_{\left\{c_{z}^{*}>0\right\}}$ is the indicator function, where the value is 1 if $c_{z}^{*}>0$ and 0 otherwise. 

### 2.4. CPU–GPU Multithreaded Pipeline Design

To make full use of GPU computational power, the aim was to design a real-time object detection system on the NVIDIA Jetson TX2, a low-power embedded heterogeneous GPU platform. Due to the low power of the TX2, energy consumption can be controlled by minimizing the calculations during system operation, and the inference speed can be improved simultaneously. Computational reduction often leads to a decline in detection accuracy, and the critical issue to be tackled is how to increase the inference speed while retaining system accuracy.

With a multicore CPU on the TX2, we maximized the computational power of the GPU via a multithreaded CPU–GPU pipeline design, where the CPU is primarily responsible for processing more logical tasks and the GPU is used to process high-density floating-point calculations. The data is transferred from the CPU memory to the GPU graphics memory. The GPU finishes processing the data for calculation and then transfers the results out to the CPU memory. 

Calculation of the detection time of the system for the object target starts with reading the image and ends with the system completing the detection and returning the result of the object and its position. Using the time detection function to count the inference time for each part of the code, we found that the time spent on the object detection process was primarily in the CPU image preprocessing and GPU network prediction stages, whereas the time for the final CPU output detection results was negligible. By further statistical analysis of the TC-YOLO network execution on the CPU and GPU, the time taken to process each frame was approximately 21 ms in single-threaded operation, with 12.6 ms executed on the GPU and 8.4 ms on the CPU. Considering that the CPU on the TX2 development board is multicore, an attempt was made to maximize the use of the computational power of the GPU by opening multiple threads for scheduling and trying to keep the GPU in constant computation. Here, one thread performs the GPU task, and another thread conducts the CPU image reading and preprocessing tasks simultaneously. When the first thread finishes the GPU computation, the second thread can immediately start the GPU computation task. The whole process carries out the GPU computation task of the previous image and the CPU preprocessing stage of the next image at the same time, so the time for preprocessing each image can be saved during the detection. Depending on the dataset and input requirements, the number of threads opened can be adjusted. We used two threads for pipelined detection depending on the current application. The final time spent on the whole process entirely hides the CPU processing time, and only the GPU processing time needs to be calculated to detect the images. In addition, the improvements proposed herein do not involve changes to the network structure and thus have no impact on the accuracy of the system.

### 2.5. Evaluation Metrics

To define the detection results in more detail, we introduced a series of evaluation metrics based on average precision (AP), including AP_50_, AP_75_, AP_S_, AP_M_, and AP_L_, where AP_50_ denotes the AP at intersection over union = 0.5, and AP_75_ indicates the AP at intersection over union = 0.75. AP_S_ indicates the AP with detection area less than 1394 (34 × 41), AP_M_ indicates the AP with detection area larger than 1394 (34 × 41) and smaller than 2888 (76 × 38), and AP_L_ refers to the AP with detection area larger than 2888 (76 × 38). The equation of AP is as follows:(11)$$\mathrm{AP}=\sum_{i=1}^{N}P\left(i\right)∆recall\;\left(i\right)$$
where *N* denotes the number of test images, *P(i)* represents the precision value at *i* images, and *recall (i)* shows the change in recall between *k* and *k* − 1 images.

#### Experimental Setup

The experiments were conducted on a server with NVIDIA Tesla V100, CUDA 11.2. The basic detection frameworks were MC-ResNetv1 and MC-ResNetv2. During training, the key hyperparameters were set as follows: learning rate = 0.0002, momentum = 0.8, gamma = 0.1, and weight decay = 0.0002. The optimizer used was stochastic gradient descent (SGD). Moreover, to ensure the test results would be more convincing, we executed the whole test process 10 times, and the final test results were averaged.

In different versions of the same model, as the input size of the image gets larger, the network needs more layers (deeper and wider) to expand the receptive fields and more channels to capture finer-grained features. Thus, the network depth or width of the backbone is typically different in various versions of the same model; in other words, their weight files are varied. If we simplistically resize the inputs of different versions to the same resolution, it would be unfair to these state-of-the-art models. To this end, we kept all parameters of the comparison models, including input size, backbone, and weights, unchanged, allowing all models to perform well. It is worth noting that the input size of the proposed model was 416 × 416 (we reshaped the 1080 × 1920 resolution to 416 × 416 resolution) to balance performance and inference speed.

## 3. Results

### 3.1. The Impact of Data Augmentation on the MC-LCNN

Data augmentation is an integral part of the whole training process and has direct impact on the final detection accuracy. We compared 14 influential data augmentation methods [15,16,17,18] and combined them to determine the final approach for dataset augmentation in this study. First, we tested the 14 enhancement methods in turn and then selected the top four performing methods to combine and test them. Typically, it is difficult for an augmentation method that performs poorly when working alone to suddenly become superior when combined with other methods, so we only considered the top four augmentation methods that perform well. The test results are shown in Table 2. 

We can clearly observe that Cutout, Blur, Flip, and Rotation achieved excellent performance with AP_50_ of 91.14%, 90.69%, 88.59%, and 88.38%, respectively. Surprisingly, the three most advanced data augmentation methods, namely Mixup, Cutmix, and Mosaic, all showed mediocre performance, probably because the image features of medicinal chrysanthemums are mostly similar, such as color, texture, etc. Thus, using complex augmentation methods can generate a large amount of redundant local information and cause overfitting. It is worth noting that the performance of Blur ranked second among all the enhancement methods, probably due to the fact that Blur makes the whole dataset increase with new features rather than redundant ones, which greatly improves the robustness of the model. Furthermore, when we combined Cutout and Blur together, the AP_50_ improved from 91.14% to 93.06%, an encouraging result. In summary, we combined Cutout and Blur as the data augmentation methods in this study.

### 3.2. Ablation Experiments

MC-LCNN employs several modules, including the proposed MC-ResNet, DropBlock, EMA, SPP, and CBM. We used ablation experiments to verify the performance of these modules. First, to validate the performance of the MC-ResNet module, we replaced MC-ResNet with 24 feature extraction networks. Furthermore, to validate the performance of DropBlock, EMA, and SPP, we removed these modules. Finally, we verified the performance of CBM by sequentially increasing the number of CBM. It is worth noting that MC-LCNN is essentially a convolutional neural network, so CBM cannot be completely removed. The results of the ablation experiments are shown in Table 3.

First, as observed in Table 3, MC-ResNet outperformed the 24 feature extraction networks, and the AP_50_, AP_S_, AP_M_, and AP_L_ were 2.13%, 3.29%, 3.14%, and 2.36% higher than the suboptimal CSPRetNeXt module, respectively, showing that MC-ResNet had the most prominent ability for small object feature extraction. Not only that, the inference speed (FPS) of MC-ResNet was an impressive 11.07% higher than that of CSPDarknet53 (the module with the second highest inference speed after MC-ResNet). Second, after adding DropBlock, EMA, and SPP, the AP_50_ of MC-LCNN improved by 3.4%, 2.12%, and 6.81% and the inference speed (FPS) improved by 2.4%, 2.59%, and 7.99%, respectively. Because SSP can receive any size of feature map input and output it to a fixed size of feature vector, this can significantly improve the detection precision and inference speed of the model. Finally, we verified that the optimal performance of MC-LCNN could be achieved using a single CBM module. When several CBM modules were employed, the AP_50_ of the whole model showed a slight increase. When using four CBM modules, the AP_50_ marginally increased by 0.27%, but the inference speed FPS significantly decreased by 19.45. To intuitively observe the image features, we show the visualization process of some images in MC-LCNN in Figure 5.

### 3.3. Comparisons with State-of-the-Art Detection Methods

In this section, we present a comprehensive comparison of the latest 13 object detection frameworks (54 models) with the proposed MC-LCNN. The results are shown in Table 4.

First, our goal was to build a lightweight network; hence, the inference speed of the model was crucial to us. The inference speed of MC-LCNN (FPS = 109.28) was second only to PP-YOLOv2 (FPS = 110.54) with an input size of 320 × 320, ranking second among the 54 models in terms of inference speed. However, the AP_50_ of MC-LCNN (93.06%) was 7.08% higher than that of PP-YOLOv2 (85.98%) with an input size of 320 × 320, showing a clear advantage. Secondly, although the inference speed of MC-LCNN was not the most superior, the detection accuracy (AP_50_ = 93.06%) was the highest among the 54 models and 3.43% higher than the suboptimal YOLOX-X (AP_50_ = 89.63%), which is an encouraging result. Not only that, in MC-LCNN, the detection precision for different anchor box sizes (AP_S_ = 69.63%, AP_M_ = 76.42%, and AP_L_ = 88.89%) was 4.41%, 2.88%, and 2.03% higher than that of the suboptimal YOLOX-X (AP_S_ = 65.22%, AP_M_ = 73.54%, and AP_L_ = 86.86%), respectively. The performance of MC-LCNN was more prominent for small-sized anchor box detection, which is critical for robotic systems that operate in natural environments. Because of path planning constraints, small-sized anchor box detection is particularly relevant when the robot picks distant chrysanthemums. Finally, according to the improvement strategy in Section 2.4, we tested MC-LCNN on a heterogeneous GPU platform, NVIDIA Jetson TX2, and the example is shown in Figure 6. 

The precision of the model remained unchanged, and the inference speed of the whole model increased by 2FPS as it benefited from the multithreaded pipeline design of the CPU–GPU. Unfortunately, we assumed that the design would completely hide CPU processing time and thus only counted GPU processing time, resulting in an improvement in detection time of approximately 19 FPS. However, due to FPS calculation and communication loss between multiple threads, the actual improvement in detection speed was different from the ideal case, although it still somewhat saved the CPU preprocessing time. The test results on the NVIDIA Jetson TX2 are shown in Figure 7.

## 4. Discussion

In response to the three issues proposed in the introduction, we have compared the proposed MC-LCNN with the studies in Table 1. For issue 1, from a detection accuracy perspective, the inference speed of MC-LCNN (9.15 ms) was slightly faster than the Liu et al. research (10 ms) [12], but the detection accuracy (AP_50_) was tremendously improved by 15.06%. From an inference speed perspective, the detection accuracy of MC-LCNN (AP_50_ = 93.06%) was 3.06% higher than the research by Yang et al. (AP_50_ = 90%) [11], with a significant improvement in inference speed from 0.7 s to 9.15 ms. MC-LCNN achieved the first highly accurate real-time testing work in the world for medicinal chrysanthemums. For issue 2, it is clear from Table 1 that most studies were tested in ideal environments or under illumination variations. In this study, the dataset was collected from natural environments, including complex unstructured environments, such as illumination variations, overlaps, and occlusions, thus significantly improving the robustness of the model. For issue 3, we tested MC-LCNN embedded in a low-power edge computing device, the NVIDIA Jetson TX2. Not only that, we used a multithreaded CPU–GPU pipeline design to improve the inference speed of MC-LCNN.

The proposed MC-LCNN has apparent advantages but also has shortcomings that need to be addressed. First, the inference speed of MC-LCNN was not optimal among all the compared models, and inference speed is crucial for robotic picking. Not only that, when the proposed model was embedded in the Jetson TX2, it took around 0.6 s to test a single image, which is an acceptable but not surprising result. Furthermore, actual unstructured environments involve more than just illumination variations, overlaps, and occlusions, and we need to collect further different scenarios to improve the robustness of the model.

## 5. Conclusions

In this work, we propose a new lightweight convolutional neural network, named MC-LCNN, for detecting medicinal chrysanthemums at the bud stage under complex unstructured environments (illumination variations, overlaps, and occlusions). We collected 4000 original images (1080 × 1920) as the dataset. In the NVIDIA Tesla V100 GPU environment, the AP_50_ of the test dataset reached 93.06%, and the inference speed was 109.28 FPS. The optimal data enhancement strategy for training MC-LCNN was the combination of Cutout and Blur. Furthermore, we compared the proposed MC-LCNN with 13 state-of-the-art object detection frameworks (54 models). MC-LCNN achieved the highest AP_50_ and was second to the optimal PP-YOLOv2 in terms of inference speed. Finally, we embedded MC-LCNN into the NVIDIA Jetson TX2 for real-time object detection and improved the inference speed by 2FPS through a multithreaded CPU–GPU pipeline design. The proposed MC-LCNN has the potential to be integrated into a selective picking robot for automatic picking of medicinal chrysanthemums via NVIDIA Jetson TX2 in the future.

## Figures and Tables

**Figure 1 plants-11-00838-f001:**
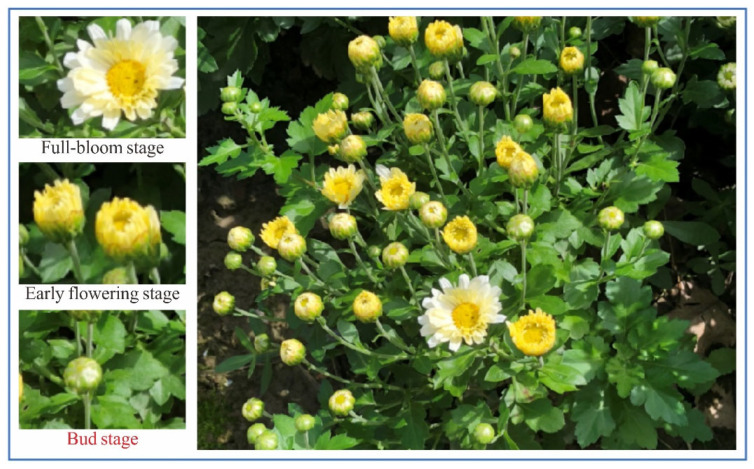
The different flowering stages of medicinal chrysanthemums.

**Figure 2 plants-11-00838-f002:**
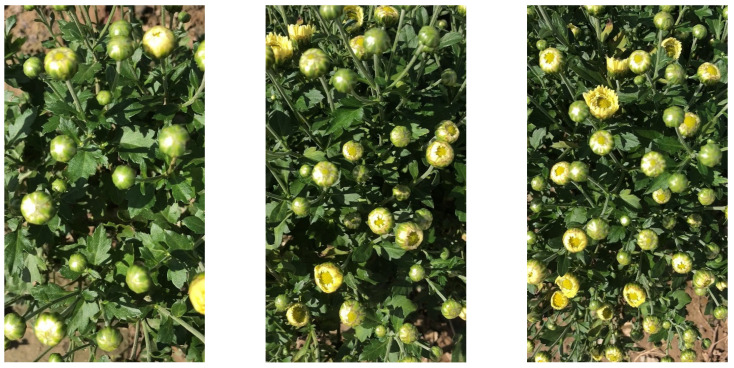
Some original images.

**Figure 3 plants-11-00838-f003:**
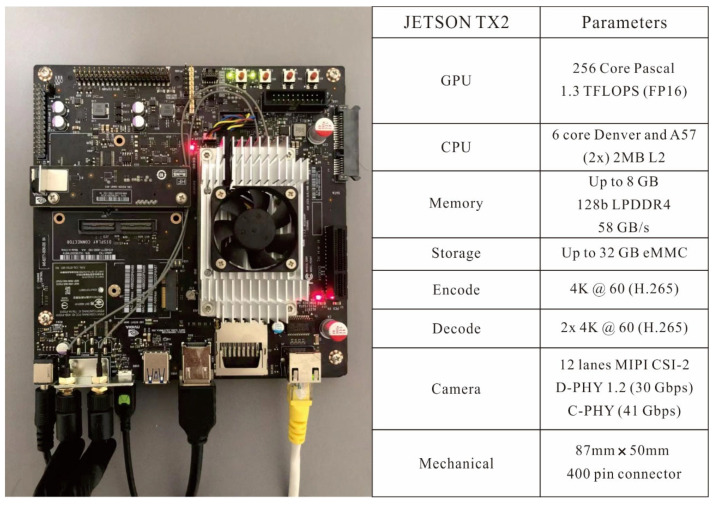
NVIDIA Jetson TX2 parameters.

**Figure 4 plants-11-00838-f004:**
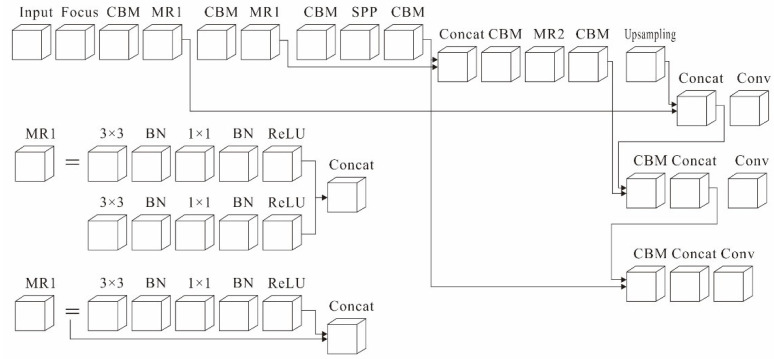
Structure of the proposed MC-LCNN.

**Figure 5 plants-11-00838-f005:**
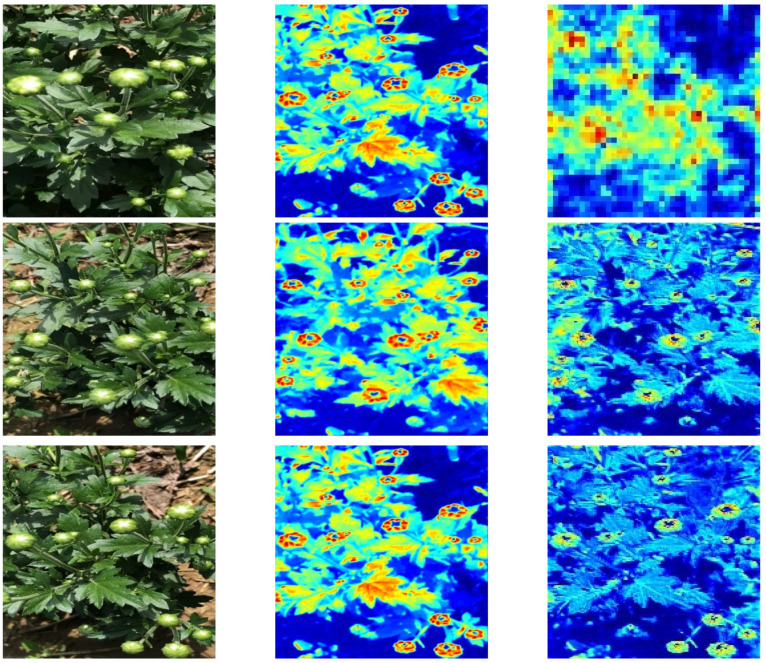
Visualization results of some input images.

**Figure 6 plants-11-00838-f006:**
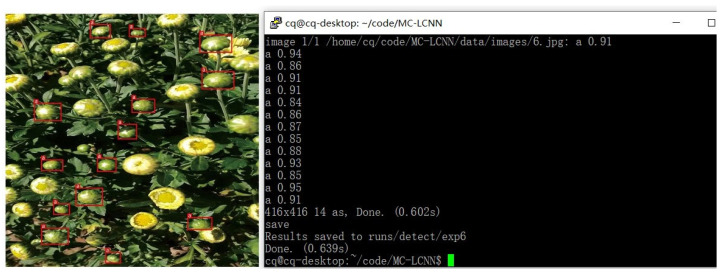
The test results on the NVIDIA Jetson TX2.

**Figure 7 plants-11-00838-f007:**
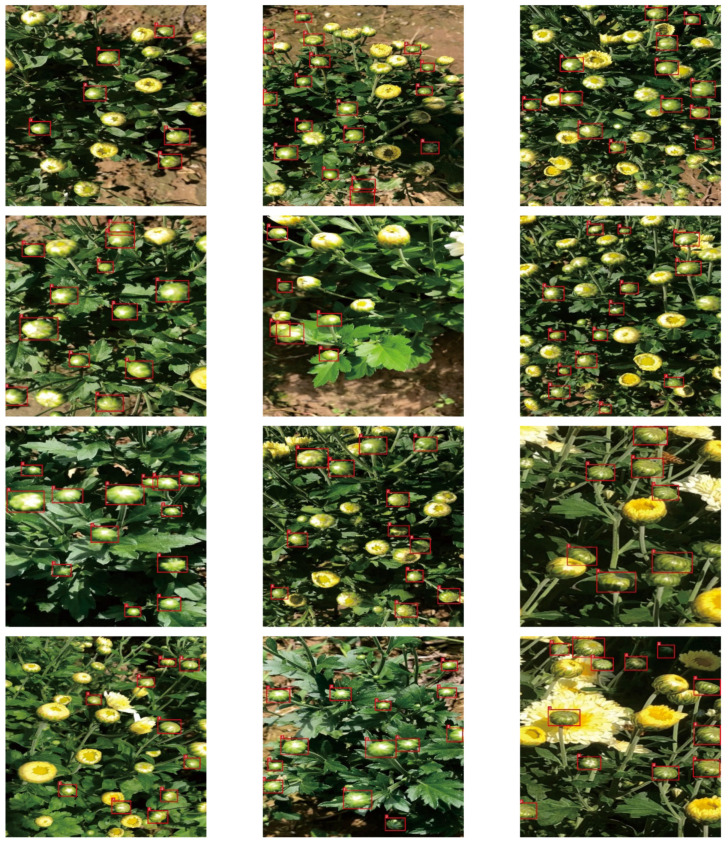
The test results on the NVIDIA Jetson TX2.

**Table 1 plants-11-00838-t001:** The literature on different chrysanthemum detection tasks.

Authors	Tasks	Published Year	Test Environment	Precision	Inference Speed	Test Devices
[6]	Chrysanthemum cut detection	1996	Ideal	/	/	Laptop
[7]	Chrysanthemum leaf recognition	2000	Ideal	/	/	Laptop
[8]	Chrysanthemum bud testing	2014	Ideal	0.75	/	Laptop
[9]	Chrysanthemum disease detection	2017	Ideal	/	/	Laptop
[10]	Chrysanthemum variety testing	2018	Illumination	0.85	0.4 s	Laptop
[11]	Chrysanthemum picking	2019	Illumination	0.9	0.7 s	Laptop
[12]	Chrysanthemum variety classification	2019	Ideal	0.78	10 ms	Laptop
[1]	Chrysanthemum variety classification	2020	Ideal	0.96	/	Laptop
[13]	Chrysanthemum image recognition	2020	Ideal	0.76	0.3 s	Laptop

**Table 2 plants-11-00838-t002:** Comparison with different data enhancement methods.

Flip	Shear	Crop	Rotation	Grayscale	Hu	Saturation	Exposure	Blur	Noise	Cutout	Mixup	Cutmix	Mosaic	AP	AP_50_	AP_75_	AP_S_	AP_M_	AP_L_
√														70.68	88.59	75.49	69.22	75.87	85.89
	√													70.99	88.63	75.32	67.01	75.22	85.28
		√												69.03	87.84	74.01	66.84	75.34	85.44
			√											69.56	88.38	74.42	66.28	76.03	85.59
				√										68.42	87.84	73.21	66.14	75.84	86.41
					√									68.82	88.44	73.57	66.11	76.03	86.04
						√								68.49	88.18	73.36	65.98	75.62	86.63
							√							69.93	89.13	73.52	66.01	75.83	86.12
								√						70.13	90.69	73.59	66.02	75.98	87.35
									√					68.06	87.11	71.25	64.39	72.88	84.11
										√				70.33	91.14	75.44	67.22	74.89	87.88
											√			68.46	88.31	72.53	65.52	73.46	85.03
												√		68.88	88.67	72.68	65.33	73.13	85.67
√			√										√	68.87	88.54	72.23	65.12	73.06	85.29
√								√						71.62	92.03	75.09	67.88	75.38	88.26
√										√				70.98	91.82	74.66	67.65	75.22	87.61
			√					√						71.44	92.36	75.93	68.66	75.87	87.99
			√							√				71.64	92.22	76.23	69.03	76.08	87.92
								√		√				72.22	93.06	76.46	69.63	76.42	88.89
√			√					√						71.88	92.62	76.32	69.12	76.53	88.22
√			√							√				71.03	92.03	75.96	68.99	75.99	87.53
			√					√		√				70.65	91.87	74.53	67.63	75.68	87.04
√			√					√		√				70.11	90.58	73.89	66.37	74.81	86.34

**Table 3 plants-11-00838-t003:** The ablation experiment results of different modules.

Method	FPS	AP	AP_50_	AP_75_	AP_S_	AP_M_	AP_L_
Ours + CBM × 5	80.29	72.44	93.31	77.02	70.23	77.39	88.93
Ours + CBM × 4	89.83	72.63	93.33	77.86	70.99	77.54	89.25
Ours + CBM × 3	96.11	72.56	93.23	77.33	70.56	77.28	89.21
Ours + CBM × 2	101.26	72.34	93.08	77.02	70.12	77.25	88.91
Ours − SPP	101.29	67.85	86.25	69.82	62.63	69.91	84.36
Ours − EMA	106.69	70.33	90.94	73.83	64.45	74.12	87.11
Ours − DropBlock	106.88	69.58	89.66	73.25	64.22	73.66	86.82
Ours (ResNet101)	64.66	64.12	85.14	66.89	58.33	67.01	82.34
Ours (ResNet50)	73.45	62.06	82.64	65.57	57.46	65.63	80.84
Ours (RetNeXt-101)	92.21	69.36	88.08	74.12	65.89	74.33	85.33
Ours (ResNet50-vd-dcn)	80.58	68.54	87.61	74.88	67.82	74.93	85.26
Ours (ResNet101-vd-dcn)	67.99	68.38	89.96	74.58	67.66	74.61	86.01
Ours (EfficientB6)	61.58	68.35	88.31	71.29	67.41	71.38	85.49
Ours (EfficientB5)	67.33	67.68	87.55	69.84	66.84	69.85	85.12
Ours (EfficientB4)	70.44	67.39	87.08	68.46	66.19	68.63	84.87
Ours (EfficientB3)	78.09	66.67	86.42	70.41	67.88	70.52	84.58
Ours (EfficientB2)	83.28	66.33	85.27	69.16	65.44	69.46	84.33
Ours (EfficientB1)	85.33	65.64	83.26	67.33	62.06	67.42	82.89
Ours (EfficientB0)	96.63	63.59	80.83	68.99	64.83	69.58	78.45
Ours (VGG16)	76.13	63.87	81.65	66.89	61.26	70.34	78.05
Ours (MobileNet v1)	83.54	62.66	79.99	72.67	66.02	72.93	76.85
Ours (MobileNet v2)	79.56	64.48	82.11	73.43	66.24	73.67	80.99
Ours (ShuffleNet v1)	85.84	65.12	84.12	69.91	61.41	70.28	82.24
Ours (ShuffleNet v2)	76.27	66.69	87.28	70.57	62.66	70.88	84.44
Ours (DenseNet)	81.02	67.34	88.54	69.66	62.16	69.99	84.83
Ours (DarkNet53)	84.82	67.98	89.67	70.18	64.53	70.22	85.06
Ours (CSPDarknet53)	98.21	68.11	89.82	72.89	65.98	72.88	85.54
Ours (CSPDenseNet)	91.46	68.14	90.22	74.33	67.38	74.56	86.22
Ours (CSPRetNeXt)	93.11	68.88	90.93	73.26	66.34	73.28	86.53
Ours (RetinaNet)	62.63	64.09	84.08	66.28	60.11	66.54	81.31
Ours (Modified CSP v5)	90.23	69.23	90.82	73.11	67.23	73.25	86.83
Ours	109.28	72.22	93.06	76.46	69.63	76.42	88.89

**Table 4 plants-11-00838-t004:** Comparisons with state-of-the-art detection methods.

Method	Backbone	Size	FPS	AP	AP50	AP75	APS	APM	APL
RetinaNet [19]	ResNet101	800 × 800	15.63	48.33	70.23	51.24	41.22	51.33	67.03
RetinaNet	ResNet50	800 × 800	18.82	51.61	76.44	55.09	44.21	55.43	69.14
RetinaNet	ResNet101	500 × 500	24.58	60.83	81.29	62.84	51.29	62.11	75.49
RetinaNet	ResNet50	500 × 500	30.99	63.69	82.99	64.44	53.09	64.13	76.58
EfficientDetD6 [20]	EfficientB6	1280 × 1280	10.26	64.13	85.21	66.45	56.33	65.91	77.27
EfficientDetD5	EfficientB5	1280 × 1280	23.58	63.09	84.66	66.31	55.94	66.35	78.21
EfficientDetD4	EfficientB4	1024 × 1024	38.61	62.99	84.33	65.11	55.31	65.36	78.01
EfficientDetD3	EfficientB3	896 × 896	50.83	60.86	83.16	64.46	54.86	64.39	77.92
EfficientDetD2	EfficientB2	768 × 768	68.99	59.54	82.84	64.08	54.11	64.12	77.87
EfficientDetD1	EfficientB1	640 × 640	80.11	56.44	79.41	58.66	49.66	58.49	72.28
EfficientDetD0	EfficientB0	512 × 512	88.29	53.28	77.96	55.86	47.26	55.89	70.21
M2Det [21]	VGG16	800 × 800	19.22	55.23	81.22	57.69	48.54	57.58	71.55
M2Det	ResNet101	320 × 320	30.54	52.33	77.38	56.54	48.44	56.36	70.83
M2Det	VGG16	512 × 512	33.56	50.19	74.94	54.46	46.21	54.32	69.91
M2Det	VGG16	300 × 300	45.44	49.68	71.86	51.33	44.37	52.68	68.58
YOLOv3 [22]	DarkNet53	608 × 608	45.31	64.65	86.85	67.23	58.57	67.66	74.83
YOLOv3(SPP)	DarkNet53	608 × 608	46.39	64.05	85.13	66.88	56.88	66.43	74.22
YOLOv3	DarkNet53	416 × 416	58.62	61.18	80.08	63.18	55.01	63.54	72.84
YOLOv3	DarkNet53	320 × 320	62.59	58.41	77.34	61.34	54.67	61.67	71.11
PFPNet (R) [23]	VGG16	512 × 512	43.11	52.22	73.59	56.24	50.88	56.68	68.42
PFPNet (R)	VGG16	320 × 320	52.09	51.35	72.63	55.12	48.89	55.37	67.95
PFPNet (s)	VGG16	300 × 300	53.64	55.53	74.33	59.81	53.22	60.44	72.67
RFBNetE	VGG16	512 × 512	36.99	60.25	80.03	62.58	54.27	62.89	75.21
RFBNet [24]	VGG16	512 × 512	52.02	58.11	76.13	61.06	53.85	61.46	75.03
RFBNet	VGG16	512 × 512	60.16	63.96	84.85	65.48	58.68	65.66	81.84
RefineDet [25]	VGG16	512 × 512	42.13	59.83	79.66	63.56	57.53	63.69	76.53
RefineDet	VGG16	448 × 448	58.61	57.51	78.09	61.11	56.91	61.41	75.54
YOLOv4 [20]	CSPDarknet53	608 × 608	49.58	66.99	88.23	69.64	60.85	69.98	86.88
YOLOv4	CSPDarknet53	512 × 512	69.42	66.38	87.98	68.99	60.44	69.33	85.34
YOLOv4	CSPDarknet53	300 × 300	83.28	63.24	83.43	66.48	59.68	66.51	80.28
YOLOv5s	CSPDenseNet	416 × 416	84.11	65.14	84.33	68.22	61.24	68.32	81.11
YOLOv5l	CSPDenseNet	416 × 416	67.03	66.35	86.26	69.31	61.37	69.41	81.33
YOLOv5m	CSPDenseNet	416 × 416	51.22	67.58	86.67	69.89	61.99	70.22	83.59
YOLOv5x	CSPDenseNet	416 × 416	30.68	68.93	88.64	72.66	63.12	72.68	84.44
PP-YOLO [26]	ResNet50-vd-dcn	320 × 320	106.85	66.64	85.26	68.15	60.85	68.17	81.23
PP-YOLO	ResNet50-vd-dcn	416 × 416	93.25	67.06	86.88	68.67	60.99	68.61	82.03
PP-YOLO	ResNet50-vd-dcn	512 × 512	80.01	68.32	87.29	69.58	61.45	69.62	83.22
PP-YOLO	ResNet50-vd-dcn	608 × 608	64.26	69.11	88.02	70.18	62.33	70.54	84.31
PP-YOLOv2 [27]	ResNet50-vd-dcn	320 × 320	110.54	67.89	85.98	68.28	62.02	68.47	82.06
PP-YOLOv2	ResNet50-vd-dcn	416 × 416	103.88	67.95	86.13	68.88	62.55	70.46	83.11
PP-YOLOv2	ResNet50-vd-dcn	512 × 512	89.04	68.36	86.85	69.33	62.84	69.67	83.89
PP-YOLOv2	ResNet50-vd-dcn	608 × 608	81.67	68.88	87.26	70.06	63.04	70.33	84.48
PP-YOLOv2	ResNet50-vd-dcn	640 × 640	63.38	69.45	88.64	71.23	64.24	71.61	85.15
PP-YOLOv2	ResNet101-vd-dcn	512 × 512	48.98	69.48	89.22	71.99	64.53	72.32	86.67
PP-YOLOv2	ResNet101-vd-dcn	640 × 640	41.34	69.66	89.59	72.83	65.11	72.88	86.88
YOLOF [28]	RetinaNet	512 × 512	102.84	65.53	86.52	69.03	62.15	69.11	83.12
YOLOF-R101	ResNet-101	512 × 512	89.28	65.91	86.58	69.44	62.41	69.45	83.48
YOLOF-X101	RetNeXt-101	512 × 512	68.09	67.56	88.34	70.95	62.95	71.06	85.66
YOLOF-X101+	RetNeXt-101	512 × 512	53.69	67.94	88.82	71.38	63.11	71.44	85.83
YOLOF-X101++	RetNeXt-101	512 × 512	36.06	68.25	89.03	72.63	64.23	72.61	86.22
YOLOX-DarkNet53	Darknet-53	640 × 640	81.61	66.89	87.41	71.12	63.28	71.29	86.13
YOLOX-M [29]	Modified CSP v5	640 × 640	65.48	67.83	88.36	71.53	63.56	71.58	86.27
YOLOX-L	Modified CSP v5	640 × 640	53.54	69.44	89.14	73.24	64.93	73.38	86.35
YOLOX-X	Modified CSP v5	640 × 640	46.22	69.86	89.63	73.39	65.22	73.54	86.86
Ours	MC-ResNet	416 × 416	109.28	72.22	93.06	76.46	69.63	76.42	88.89

## Data Availability

Not applicable.

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
