# Peer review of "Medicinal Chrysanthemum Detection under Complex Environments Using the MC-LCNN Model"

_plants, 2022, doi:10.3390/plants11070838_

Round 1

Reviewer 1 Report

The subject of the article is interesting, and it is linked to the objectives of the journal, however, there are some issues that have to be reconsidered.

For better visibility on databases, the authors are asked not to repeat among keywords the words/concepts included in the title of the article. 

Some of the pictures seem useless to me (e.g. Figure 3, even Fig 1, as on Fig 5 there are included some original inputs)

Figure 4. Please indicate if you are the author of it

All in all it seems to me an interesting and well-written article.

Reviewer 2 Report

Recommendations to authors

  • Line 14: Please consider replacing “desirable” with “useful or crucial”
  • Line 400, Table 4: Shouldn’t all the state-of-the-art detection methods use the same resolution to be compared under the same conditions? Please use the same image resolution for all the methods under consideration or otherwise you must explain why you did not choose the same resolution for all. Why the resolution is not 1080 × 1920 (the resolution of the original photos taken)? Please explain in the text.
  • Line 416: Please replace “We” with “we”.
  • Line 478: Please provide more information about the "natural environment" mentioned in the text. Have you also checked the performance of the proposed methodology during a cloudy day? Are there any weather / lighting restrictions?
  • Line 492: Consider adding a limitation section to indicate what conditions must be met by the proposed methodology for best results in detecting medicinal chrysanthemums.
